# Plasma microRNAs Associate Positive, Negative, and Cognitive Symptoms with Inflammation in Schizophrenia

**DOI:** 10.3390/ijms252413522

**Published:** 2024-12-17

**Authors:** Takuya Miyano, Masakazu Hirouchi, Naoki Yoshimura, Kotaro Hattori, Tsuyoshi Mikkaichi, Naoki Kiyosawa

**Affiliations:** 1Translational Science Department II, Daiichi Sankyo Co., Ltd., 1-2-58 Hiromachi, Shinagawa, Tokyo 140-8710, Japan; masakazu.hirouchi@daiichisankyo.com (M.H.); tsuyoshi.mikkaichi@daiichisankyo.com (T.M.); naoki.kiyosawa@daiichisankyo.com (N.K.); 2Department of Psychiatry, National Center Hospital, National Center of Neurology and Psychiatry, Tokyo 187-8551, Japan; naoyoshi@ncnp.go.jp; 3Department of Bioresources, Medical Genome Center, National Center of Neurology and Psychiatry, Tokyo 187-8551, Japan; hattori@ncnp.go.jp

**Keywords:** circulating microRNA, schizophrenia, patient subgroups, clinical biomarker, inflammation

## Abstract

Schizophrenia is a complex and heterogenous psychiatric disorder characterized by positive, negative, and cognitive symptoms. Our previous study identified three subgroups of schizophrenia patients based on plasma microRNA (miRNA) profiles. The present study aims to (1) verify the reproducibility of the miRNA-based patient stratification and (2) explore the pathophysiological pathways linked to the symptoms using plasma miRNAs. We measured levels of 376 miRNAs in plasma samples of schizophrenia patients and obtained their Positive and Negative Syndrome Scale (PANSS) scores and the Brief Assessment of Cognition in Schizophrenia (BACS) scores. The plasma miRNA profiles identified similar subgroups of patients as in the previous study, suggesting miRNA-based patient stratification is potentially reproducible. Our multivariate analysis identified optimal combinations of miRNAs to estimate the PANSS positive and negative subscales and BACS composite scores. Those miRNAs consistently enriched ‘inflammation’ and ‘NFκB1′ according to miRNA set enrichment analysis. Our literature-based text mining and survey confirmed that those miRNAs were associated with IL-1β, IL-6, and TNFα, suggesting that exacerbated positive, negative, and cognitive symptoms are associated with high inflammation. In conclusion, miRNAs are a potential biomarker to identify patient subgroups reflecting pathophysiological conditions and to investigate symptom-related molecular mechanisms in schizophrenia.

## 1. Introduction

Schizophrenia is a complex and heterogenous psychiatric disorder characterized by positive (e.g., hallucinations), negative (e.g., emotional withdrawal), and cognitive (e.g., deficits in working memory) symptoms. These symptoms exhibit diverse manifestations across patients, with temporal fluctuations in individual cases [1,2,3]. A standardized measurement of the symptoms is essential to perform appropriate assessment and to offer optimal treatment for each patient [4]. Examples of those measurements are the Positive and Negative Syndrome Scale (PANSS), which is a clinician-rated scale for positive and negative symptoms and general psychopathology [5], and the Brief Assessment of Cognition in Schizophrenia (BACS), which is a concise cognitive battery comprising six subtests, evaluating verbal memory, working memory, motor speed, verbal fluency, attention, and executive function [6]. These standardized measurements have delivered insights into the patient subgroups based on symptoms [7] and different response patterns to antipsychotic treatments [8].

Significant challenges remain in treating those symptoms due to the heterogeneous response to antipsychotic treatments. While antipsychotic therapy is available for treating the positive symptoms, approximately 30% of patients experience persistent positive symptoms despite adequate treatment using multiple antipsychotic medications, a condition known as treatment-resistant schizophrenia [9]. Moreover, current antipsychotic treatments have limited efficacy on the negative and cognitive symptoms [2,10]. The prevalence of poor responders to antipsychotics and the challenges in developing new drugs for the poor responders can be attributed, at least in part, to an insufficient understanding of the heterogeneous pathophysiology of schizophrenia. Elucidating this heterogeneity at the molecular level is crucial to providing optimal treatments for each patient and to facilitating the development of novel drugs.

Omics-based molecular biomarkers have been explored to understand the heterogenous pathophysiology of schizophrenia. Transcriptome analyses of post-mortem brains have elucidated subgroups of schizophrenia patients, suggesting γ-aminobutyric acid [11], immune-related [12] and proteasome-related [13] pathways as underlying mechanisms for heterogeneity. Proteome analysis of serum samples has identified two subgroups of schizophrenia patients: one characterized by immune molecules and the other by growth factors and hormones [14].

The heterogeneity of the pathophysiology may reflect that of the symptom. A transcriptome analysis of whole blood samples has revealed that immune-related genes correlated with PANSS positive subscales, while mitochondrial pathway-related genes correlated with PANSS negative subscales [15]. A proteome analysis of serum samples identified several proteins (e.g., C-reactive protein and osteopontin) that correlated with PANSS positive and negative subscales [16]. These transcriptome and proteome data provide valuable information for understanding the molecular mechanisms underlying the heterogenous pathophysiology within schizophrenia. However, their potential as clinical biomarkers is limited by their instability under storage conditions because both mRNAs and proteins are prone to degradation during the storage and processing, thereby potentially affecting their reliability [17,18].

Circulating microRNAs (miRNAs), such as plasma miRNAs, have attractive features as clinical biomarkers [19]. miRNAs exhibit exceptional post-sampling stability due to the resistance to endogenous RNase activity, even under harsh conditions regarding temperatures, pH, storage periods, and freeze–thaw cycles [20,21]. Circulating miRNAs offer a less invasive alternative to brain tissue sampling and may serve as potential indicators of cerebral biological processes because brain-derived miRNAs can pass through the blood–brain barrier via exosomes and enter the systemic circulation [22]. On the other hand, miRNAs are also found in protein complexes and apoptotic bodies in body fluids [23], originating from various tissues and cellular processes throughout the body. In addition, miRNAs have the potential to capture a broad spectrum of biological information because each miRNA modulates the post-transcriptional expression of multiple target genes, thereby influencing diverse physiological processes such as inflammation and neuroplasticity [24].

Circulating miRNAs have revealed several aspects of heterogeneity among schizophrenia patients. Pérez-Rodríguez et al. discriminated treatment-resistant schizophrenia from antipsychotic treatment responders among schizophrenia patients using a whole blood miRNAs signature [25]. Lai et al. revealed several miRNAs in peripheral mononuclear leukocytes were differentially correlated with PANSS negative subscales and neurocognitive performance scores [26].

Our previous study identified three subgroups of schizophrenia patients based on plasma miRNA profiles, where the subgroups were characterized by different inflammatory backgrounds [27]. One major limitation in the previous study was the small size of the patient sample (*n* = 26). Thus, the reproducibility of these findings remains to be established.

The present study aims to (1) verify the reproducibility of the miRNA-based patient stratification and (2) explore pathophysiological pathways linked to symptoms using plasma miRNAs. The present study demonstrates that plasma miRNAs reproducibly stratified schizophrenia patients into three subgroups in an independent cohort. Furthermore, our miRNA analysis associated the positive, negative, and cognitive symptom scores with inflammation. These results suggest that miRNAs are promising clinical biomarkers for identifying patient subgroups and bridging symptoms with pathophysiological signals. These findings may contribute to realizing precision medicine and developing novel therapeutic strategies.

## 2. Results

### 2.1. Plasma miRNA Profiles Identified Similar Subgroups of Schizophrenia Patients as in the Previous Study

We tested if plasma miRNA profiles can identify three subgroups of schizophrenia patients as in the previous study [27], using an independent cohort of patients. We measured expression levels of 376 miRNAs in plasma of 70 schizophrenia patients (Table 1). The whole miRNA profiles identified three subgroups (*a*, *b*, and *c*) of the patients (Figure 1A), where subgroup *a* exhibits low miRNA levels in the second quadrant from the left, subgroup *b* shows high miRNA levels in the left half, and subgroup *c* has low miRNA levels in the rightmost third. There was no apparent bias in age and sex between the subgroups (Appendix A).

We investigated the similarity between the miRNA-based subgroups in this study and those in the previous study [27]. We visualized a heatmap of the miRNA levels in this study using the predefined miRNAs that were distinctive among the subgroups in the previous study (Figure 1B). The predefined miRNAs identified three patient subgroups (1′, 2′, and 3′) with miRNA patterns similar to those in the previous study. The subgroups based on the predefined miRNAs were largely matched with those based on the whole miRNA profiles (26 out of 32 patients in the subgroup 1′ were clustered into subgroup *b*, 19 out of 33 patients in the subgroup 2′ were clustered into subgroup *c*, and 4 out of 5 patients in subgroup 3′ were clustered into the subgroup *a*). There was no apparent bias in the age and sex between the subgroups (Appendix A). Although the miRNA-based subgroups were similar between the present and previous studies, subgroup 2′ in this study did not exhibit the high PANSS levels, whereas subgroup 2 in the previous study showed relatively high PANSS levels.

### 2.2. Multivariate Analysis Identified Symptom-Related miRNAs as Optimal Combinations of miRNAs to Estimate Symptom Scores

We first applied a univariate analysis to explore the associations between miRNA levels and the symptom scores. We calculated correlation coefficients for all the combinations between the 376 miRNAs and the symptom scores (i.e., PANSS total scores, PANSS positive subscales, PANSS negative subscales, PANSS general psychopathy subscales, and BACS composite scores) (Table 2 and Appendix A). No single combination demonstrated a significant correlation (Benjamini–Hochberg corrected *p* >0.05).

We then conducted a multivariate analysis to explore the associations between miRNA levels and symptom scores because multiple miRNAs have potential to reveal more complex relationships with symptoms than a single miRNA [28]. We developed multivariate regression models based on partial least squares (PLS) to estimate the symptom scores using miRNA levels. In the model development process, we optimized the combinations of miRNAs as input variables through a forward stepwise method to minimize the root mean standard error of cross validation (RMSECV), which is the estimation error in leave-one-out cross validation (Figure 2A,B and Appendix A). The estimation accuracy of the models with the optimal miRNAs was confirmed via a scatter plot between observed and estimated symptom scores in the cross validation (Figure 2C) and their RMSECV: 2.786 for PANSS positive subscales (range: 7–38), 2.359 for PANSS negative subscales (range: 7–36), 4.961 for PANSS general psychopathy subscales (range: 16–65), 6.696 for PANSS total scores (30–120), and 0.308 for BACS composite scores (range: −4.467–0.476). There was no notable influence of time difference from blood sampling to PANSS/BACS measurement, age, and sex on the estimation accuracy in the cross validation (Appendix A). We regarded the optimal combination of miRNAs to estimate symptom scores as symptom-related miRNAs. Since the cross validation generates different models for each sample, the final model was calibrated using all the samples. Regression coefficients of the final models can indicate the impact of the optimized miRNA on the symptom scores (Appendix A). For example, negative regression coefficients indicate an inverse relationship between miRNA levels and symptom scores, suggesting that higher levels of specific miRNAs are associated with lower symptom scores.

### 2.3. Symptom-Related miRNAs Enriched Inflammatory Pathways and Were Reported to Regulate IL-1β, IL-6, and TNFα

We explored pathological pathways linked to the symptoms through the miRNAs. We performed miRNA set enrichment analysis on the symptom-associated miRNAs (Table 3). ‘Inflammation’ and ‘NFκB’ were enriched in all the miRNA sets except for PANSS general psychopathy, which did not enrich any function. In addition, brain-related terms were enriched in all the miRNA sets, except for PANSS general psychopathy (‘brain.cerebellum’ in PANSS positive-related miRNAs, ‘brain’ in PANSS negative-related miRNAs and PANSS total-related miRNAs, and ‘brain.nucleus_caudatus’ in BACS composite-related miRNAs). The subsequent analysis examining the association between the symptoms and inflammation focused on PANSS positive subscale, PANSS negative subscale, PANSS total scores, and BACS composite scores, while excluding PANSS general psychopathology, as it did not enrich ‘inflammation’.

We consolidated the association of the symptom-related miRNAs with inflammation based on literature information. We conducted literature-based text mining to quantify the publications that associate miRNAs with diseases or proteins. Among disease-related terms, ‘inflammation’ was reported as one of the top two most frequently associated with the miRNAs for all the evaluated symptoms (i.e., PANSS positive and negative subscales, PANSS total scores, and BACS composite scores) (Figure 3A). Similarly, among protein-related terms, the pro-inflammatory cytokines IL-1β, IL-6, and TNF-α, as well as other proteins, including TGF-β1, PTEN, BCL2, VEGF-A, AKT1, STAT3, CASP3, and MYC, were consistently reported as one of the top twenty most frequently associated with the miRNAs (Figure 3B).

We investigated whether the symptom-related miRNAs can associate the deterioration of the symptoms with increased levels of IL-1β, IL-6, and TNFα. We surveyed studies that experimentally demonstrate the regulatory functions of the symptom-related miRNAs on IL-1β, IL-6, and TNFα. We identified studies that experimentally demonstrate the miRNAs positively or negatively regulated with IL-1β, IL-6, and TNFα (Appendix A). We then qualitatively inferred the levels of cytokines in each subgroup (Figure 4) based on the experimental evidence and the regression coefficients of the multivariate regression models. Some miRNAs with positive regression coefficients in the models estimating PANSS positive and negative subscales and PANSS total scores were reported to upregulate IL-1β, IL-6, and TNFα, while others with negative regression coefficients were reported to downregulate these cytokines. These results suggest that patients with high PANSS scores (positive, negative, and total scores) could have high levels of IL-1β, IL-6, and TNFα. For example, PANSS positive subscales were associated with high levels of miR-9 according to the regression coefficients in the PLS models (Appendix A). The miR-9 was reported to upregulate IL-1β (Appendix A). Therefore, high PANSS positive subscales were associated with high levels of IL-1β (Figure 4). Similarly, some miRNAs with positive regression coefficients in the model estimating BACS composite scores were reported to downregulate IL-1β, IL-6, and TNFα, while others with negative regression coefficients were reported to upregulate these cytokines. These results suggest that patients with low BACS composite scores could have high levels of IL-1β, IL-6, and TNFα.

## 3. Discussion

### 3.1. Plasma miRNA Profiles Are Potentially Reproducible Biomarkers to Stratify Schizophrenia Patients

The present study reproducibly demonstrated the potential of plasma miRNAs to stratify schizophrenia patients into three subgroups, consistent with findings from previous research [27]. The overall miRNA profiles, which are the measured miRNAs in each study, identified three subgroups of schizophrenia patients in both the present (Figure 1A and Table 1) and previous [27] studies. The predefined miRNAs that distinguished the subgroups in the previous study also identified three patient subgroups in the present study with similar miRNA patterns (Figure 1B), suggesting that the predefined miRNAs may serve as a signature to identify comparable patient subgroups across different datasets. Our findings demonstrated that both overall miRNAs and the predefined miRNAs identified similar patient subgroups (Figure 1B). These findings suggest that our study identified comparable patient subgroups to those in the previous study, even without relying on the predefined miRNAs.

The miRNA-based identification of three patient subgroups may be robust to the differences in the data backgrounds between the previous and present studies. One major difference lies in the patients’ backgrounds. Races of the patients in the previous study were ‘Black or African-America’ and White, while those in the present study were Japanese. Additionally, the phases of symptoms were different; the previous study included acute conditions only while the present study did not set specific inclusion criteria on the phase of symptoms. There was also a notable difference in the severity of symptoms: mean PANSS total scores of 93.3 ± 9.5 in the previous study and 68.4 ± 20.7 in the present study. Furthermore, the medication protocols varied; a washout period for antipsychotics was implemented in the previous study, whereas antipsychotic use was allowed in the present study. The study designs also differed; the previous study recruited patients in a clinical trial, while the present study included patients receiving routine medical care. Another difference lies in the miRNA measurement platform; the previous study measured 179 miRNAs, while the present study analyzed 376 miRNAs, with 136 miRNAs shared between the two studies. Despite these differences, the plasma miRNA profiles reproducibly identified three patient subgroups, suggesting that miRNA-based patient stratification is robust against such variations across different studies.

Meanwhile, some findings were inconsistent between the present and previous studies. The present study did not associate the miRNA-based subgroups with distinctive PANSS profiles, whereas the previous study demonstrated higher PANSS scores in the miRNA-based subgroup 2. This discrepancy may be attributed to either (1) a false-positive result in the previous study due to its limited sample size (*n* = 26) or (2) a true-positive finding that lacks robustness across the different data backgrounds mentioned above. Either way, this discrepancy suggests that PANSS profiles are not robust surrogates for the miRNA-based patient stratification, although the previous study suggested that both miRNAs and PANSS profiles could be a potential biomarker to identify the patient subgroups. Thus, miRNAs may be a more reliable biomarker than PANSS to identify patient subgroups across the different datasets. Another discrepancy is that the present study demonstrated less distinctive patterns of the predefined miRNAs compared with the previous study (Figure 1B). This is presumably because the predefined miRNA was defined using the data in the previous study only, and, thus, there is limited extrapolation capability for the data in the present study. There is room for further investigation to determine which miRNAs are optimal as signatures to obtain more reproducible results on the patient stratification across different datasets.

### 3.2. Plasma miRNAs Are a Potential Molecular Biomarker to Reflect Severities of Positive, Negative, and Cognitive Symptoms

We presented a simple yet effective approach to identify symptom-related miRNAs by combining PLS regression, leave-one-out cross validation, and forward stepwise variable selection (Figure 2A,B). PLS is a linear regression technique that can handle multicollinearity, thereby enabling the use of a large number of variables (e.g., 376 miRNAs), even with small sample sizes (e.g., 41 samples) while avoiding overfitting [29]. Leave-one-out cross validation utilizes data from limited sample sizes by using all samples in both training and validation, while maximizing the number of training samples in each iteration of cross validation [30]. Forward stepwise variable selection efficiently explores optimal combinations of miRNAs with reduced computational cost by starting with a small number of miRNAs and incrementally adding a beneficial miRNA to improve the estimation accuracy. Their combination provides a framework to identify the optimal set of miRNAs in estimating symptom scores. Each method can be tailored to the specific characteristics of datasets. PLS may not be suitable to handle non-linear associations between miRNAs and symptom scores. In such instances, non-linear alternatives like kernel PLS may be more appropriate, although they involve higher computational demands [31]. Leave-one-out cross validation can be computationally intensive and may lead to high variance in model assessment. Instead, *k*-fold cross validation may provide more computationally efficient and stable results, although it can have more bias in the model assessment [32]. When forward stepwise variable selection fails to identify miRNAs with sufficient estimation accuracy, alternative techniques such as Variable Importance in Projection [33] can be employed, or variable selection methods can be tailored to the specific characteristics of the data [34].

Our multivariate analysis demonstrated the potential of plasma miRNAs to estimate the symptom scores. While multiple studies have reported correlations between individual miRNA expression levels and symptom scores in schizophrenia using univariate analysis, these correlations often lack reproducibility across different studies [35,36]. To address this issue, we propose analyzing multiple miRNAs as a signature. This approach may uncover associations that are not detectable when examining each miRNA independently. Our PLS models using optimal miRNAs successfully estimated positive, negative, and cognitive symptom scores in leave-one-out cross validation (Figure 2C), despite no single miRNA significantly correlating with symptom scores in our univariate analysis (Table 2 and Appendix A). This miRNA signature approach has shown promise in other contexts, such as discriminating schizophrenia patients from healthy controls [26] and identifying treatment-resistant schizophrenia [25]. Given these results, miRNA signature-based approaches can be valuable for associating miRNAs with symptom scores in schizophrenia, potentially offering improved reproducibility and robustness compared with univariate methods.

Plasma miRNAs are potential clinical biomarkers to deliver multifaceted information including, but not limited to, symptom severity. The miRNA-based estimation of symptom scores may serve as an objective surrogate for subjective symptom evaluations, which can be time consuming, as in cognitive function assessments [37], thereby reducing the burden on patients and clinicians. Additionally, miRNA-based prediction of responsiveness to antipsychotics, as demonstrated by Pérez-Rodríguez et al. [25], may contribute to the design of individualized treatment plans. MicroRNAs can also provide insights into other clinically relevant factors, such as potential responses to investigational drugs and risk of adverse effects, provided that miRNAs–factor relationships are established. Given the cumulative nature of miRNA-associated information, a single measurement of plasma miRNAs has potential to provide multifaceted information for precision psychiatry. Further exploration of the associations of miRNAs with clinical and pathophysiological parameters is essential to expand the utility of miRNAs in psychiatric practice.

### 3.3. Exacerbated Positive, Negative, and Cognitive Symptoms Are Associated with High Inflammation

Our analysis on the symptom-related miRNAs suggested inflammation as a common pathway for pathophysiology underlying the positive, negative, and cognitive symptoms. The miRNA set enrichment analysis on the symptom-related miRNAs for PANSS positive subscales, PANSS negative subscales, and BACS composite scores consistently enriched ‘inflammation’ and ‘NFκB1′ (Table 3). NFκB is a master immune transcription factor, which regulates inflammatory cytokines, including IL-1β, IL-6, and TNFα, and is involved in the pathogenesis of schizophrenia [38,39]. These results suggest that NFκB-related inflammatory pathways such as IL-1β, IL-6, and TNFα are associated with the symptom-related miRNAs.

Our literature-based text mining and survey on the symptom-related miRNAs associated exacerbated symptoms with high levels of inflammation. The text mining analysis revealed that the symptom-related miRNAs have been frequently associated with inflammation, IL-1β, IL-6, and TNFα in the literature (Figure 3). In addition, our literature survey on experimental evidence confirmed that some of those symptom-related miRNAs were reported to upregulate or downregulate IL-1β, IL-6, and TNFα (Appendix A). Logical inference combining this literature-based experimental evidence with the regression coefficients of the models estimating the symptom scores (Appendix A) suggested that patients with severe symptom scores have high levels of IL-1β, IL-6, and TNFα (Figure 4).

Consistent with those findings in our miRNA-based analysis, precedented research has associated schizophrenia symptoms with inflammation. The severity of positive symptoms correlated with plasma IL-6 [40] and IL-1β [41] levels. The severity of negative symptoms correlated with serum IL-1β [42,43] and IL-6 [43,44] levels and plasma TNFα levels [42,43,45]. The severity of cognitive impairment correlated with plasma IL-1β [46] and TNFα levels [46] and serum IL-6 levels [47]. Symptom-associated inflammation is not limited to these three cytokines. For example, our text mining analysis associated the symptom-related miRNAs with TGF-β1 consistently for the PANSS positive, negative, total scores, and BACS composite scores (Figure 3B). A study reported that plasma TGF-β1 levels were correlated with PANSS total scores [48]. Thus, TGF-β1 may also be involved in the inflammation symptom association, although we focused on IL-1β, IL-6, and TNFα as representative pro-inflammatory cytokines.

Our text mining analysis of the symptom-related miRNAs may provide deeper insights into the pathophysiology of schizophrenia, extending beyond inflammation. By quantifying publications that associate these miRNAs with various diseases and proteins across approximately 1850 journals, we identified numerous associations, not limited to inflammation and cytokines. Further investigation into the schizophrenia’s symptom-associated disease/proteins may provide valuable insights into the pathophysiology of schizophrenia.

Osteoarthritis, a non-inflammatory joint disorder, was among the most frequently reported diseases associated with symptom-related miRNAs (Figure 3A). This association through miRNAs implies that schizophrenia and osteoarthritis may share common pathophysiological mechanisms. The potential relevance between the two diseases is supported by epidemiological and genetic risk studies, which have demonstrated that schizophrenia patients and individuals with high schizophrenia polygenic risk scores had a reduced risk of osteoarthritis [49,50,51]. While osteoarthritis is not typically classified as an inflammatory arthritis like rheumatoid arthritis, inflammatory cytokines such as IL-1β, IL-6, and TNFα play a role in its pathogenesis [52]. Specifically, TNF-α is responsible for not only inflammatory pain but also osteoclast proliferation and differentiation, which are closely related to joint pain [53]. Hence, dysregulated inflammatory cytokines may potentially be a common pathophysiology for schizophrenia and osteoarthritis. Further investigation into such common pathways between schizophrenia and osteoarthritis may give a clue to enhance understandings on the pathogenesis of both diseases.

PTEN, which is a tumor suppressor gene and inhibits the PI3K/AKT pathway [54], was among the most frequently reported proteins associated with the symptom-related miRNAs (Figure 3B). The association between PTEN and schizophrenia via miRNAs implies that PTEN is involved in the pathophysiological mechanisms of the symptoms. Consistent with our findings, research has associated the PTEN/PI3K/AKT axis with the pathogenesis of schizophrenia [55,56]. For instance, genetic variants regarding PI3K/AKT were associated with the risk of schizophrenia, suggesting PI3K/AKT is one of the causal mechanisms of schizophrenia [57,58]. In addition, antipsychotics have ameliorated schizophrenia-like behavior through a mechanism dependent on the PI3K/AKT axis in a rat model [59]. Our miRNA-based findings corroborate the PTEN/PI3K/AKT axis as one of the key pathways of schizophrenia and, thus, a potential target for schizophrenia treatment. In addition to schizophrenia treatment, antipsychotics have demonstrated anti-glioblastoma activity by stabilizing PTEN, suggesting that the PTEN/PI3K/AKT axis is also a potential target for glioblastoma [60]. Collectively, miRNA research in schizophrenia has potential to expand insights into the pathophysiology and treatments of other diseases as well as schizophrenia.

### 3.4. Anti-Inflammatory Treatments Are Potentially Effective for Positive, Negative, and Cognitive Symptoms in a Specific Population

Anti-inflammatory drugs have demonstrated their potential to improve positive, negative, and cognitive symptoms. Cho et al. conducted a meta-analysis of adjuvant anti-inflammatory drugs [61]. Their meta-analysis demonstrated that, overall, anti-inflammatory agents significantly reduced PANSS total, positive, and negative symptom scores, and minocycline and pregnenolone significantly improved cognitive symptoms. They also revealed that the effects of anti-inflammatory agents for the positive symptoms were enhanced in patients with high total PANSS scores at baseline. Their subgroup analysis especially indicated that aspirin was more effective for clinical trials with high PANSS total scores for the recruited patients, suggesting that there is heterogeneity among schizophrenia patients in terms of responsiveness to the anti-inflammatory treatments as well as antipsychotics.

Patients with high inflammatory backgrounds might demonstrate enhanced therapeutic benefits from anti-inflammatory drugs. As discussed in the previous section, both our miRNA-based findings and existing research indicate that exacerbated symptoms can be associated with high levels of inflammation. These observations suggest that individuals with more severe symptoms, which may reflect higher inflammatory states, could be more responsive to anti-inflammatory drugs. Additionally, high levels of inflammation have been linked to treatment-resistant schizophrenia [62]. Thus, anti-inflammatory treatments may offer potential benefits for those patients. To clinically verify this hypothesis and optimize treatment strategies, it is essential to identify such high-inflammation patients using reliable clinical biomarkers.

Plasma miRNAs are potential biomarkers to identify patients with high inflammation, who may benefit from anti-inflammatory treatments. Two strategies can be considered for using miRNAs to identify such patients. The first approach is to identify patient subgroups based on miRNA profiles and select specific subgroups. Our previous study revealed that subgroups 1 and 2 had higher inflammation; therefore, anti-inflammatory drugs might be effective for patients in subgroups 1 and 2, corresponding to subgroups 1′ and 2′ in the present study (Figure 1B). The second approach is to select patients who are estimated to have high symptom scores based on miRNA profiles. The miRNAs adopted in our PLS model are expected to reflect the levels of inflammatory cytokines (Figure 4). Consequently, selecting patients with high estimated symptom scores may indirectly identify those with high inflammation. Given the advantageous features of plasma miRNAs as clinical biomarkers, including their high stability after blood sampling, miRNAs represent a promising approach to identify patients with high inflammation. This strategy potentially enhances the efficacy of anti-inflammatory drugs. The plasma miRNA-based patient stratification may lead to the realization of personalized therapeutic interventions for schizophrenia.

### 3.5. Limitations of This Study Inform Directions for Future Research

The sample size and diversity of patients in this study may still be insufficient to generalize our findings. While plasma miRNA profiles reproducibly identified three subgroups of patients, we were unable to evaluate the influence of demographic factors such as ethnicity or symptom scores on the subgroup identification. Regarding the estimation of symptom scores using plasma miRNA levels, we could only assess estimation accuracy through cross validation, and we cannot rule out the possibility that the selected miRNAs may be overfitted to this particular dataset. It is necessary to evaluate its performance on an independent test set to determine the true generalizability of our models. Additionally, this study analyzed miRNA expression at only one time point per subject. Longitudinal evaluations could provide insights into whether these profiles indicate stable traits or fluctuating states in individuals. Increasing the sample size and including a more diverse patient population would enhance the robustness and generalizability of the subgroup identifications and the symptom score estimation.

The patient subgroups were identified based on the relative levels of miRNAs among patients in each study, as visualized in the heatmap of miRNA profiles (Figure 1). However, due to the use of different miRNA measurement platforms, the miRNA levels and color scales in the present study are not directly comparable to those in the previous study [27]. Consequently, it is uncertain whether subgroup 2 in the previous study and subgroup 2′ in the present study represent the same patient population. With a view to develop a diagnostic biomarker to determine which subgroup each patient belongs to in clinical practice, it would require an approach that enables assessment on a patient-by-patient basis. To achieve this, it would be essential to establish a consistent and robust measurement platform along with an appropriate discrimination model. Measurement platforms need to be validated for the intended use, e.g., as clinical biomarkers in clinical practice, because they have different reproducibility, bias, specificity, sensitivity, and accuracy [63,64]. A discrimination model may, for example, calculate a composite score based on a miRNA signature and set a threshold to determine whether an individual patient belongs to a specific subgroup.

The patient population in this study reflects the diverse patient demographics encountered in routine clinical practice, rather than adhering to strict inclusion/exclusion criteria typical of clinical trials. Several factors were not controlled, including treatment history with antipsychotics, dietary habits, and smoking status. To evaluate the potential impact of these uncontrolled variables on miRNA-based patient stratification and symptom severity estimation, future investigations would require appropriately controlled patient cohorts with standardized protocols for confounding factor management.

Our approach to interpreting miRNA functions combines three literature-based methods (i.e., miRNA set enrichment, text mining, and manual literature survey) but is still not exhaustive. The miRNA set enrichment analysis tool TAM2.0 [65], which utilizes a literature-derived database of miRNA–pathway associations, has not been updated since 2020, thus failing to reflect recent research findings. The text mining tool EmBiology, used to quantify miRNA-protein/disease associations, has several limitations. Its full-text search was restricted to approximately 1850 journals, with half from Elsevier B.V. and half from other publishers. Moreover, its text-based associations do not guarantee experimental validation. In addition, the simple enumeration of literature reports without normalizing for pathway frequency across all journals may overrepresent commonly studied pathways such as inflammation. Our manual literature survey, which explored experimental evidence of miRNA effects on cytokines, cannot ensure comprehensive journal coverage due to the labor-intensive nature of the literature search and perusal of experimental conditions and results. Additionally, we focused on the experimental evidence supporting the hypothesis that exacerbated symptoms are associated with high inflammation. However, there could exist conflicting evidence for functions of miRNAs on cytokines. For instance, one study demonstrated miR-16 mimic upregulated TNFα in HT29 cells [66] (Appendix A), while another found it downregulated TNFα in NH7A cells [67]. An automatic and objective survey would enable us to compare such conflicting evidence comprehensively, thereby determining which direction of regulatory function (e.g., whether miR-16 upregulates or downregulates TNFα) is more plausible. The development of an artificial intelligence-based literature search tool capable of analyzing both text and figures from a broader range of journals would enhance the comprehensiveness and objectivity of miRNA functional analysis.

While this study focused on miRNAs, other small non-coding RNAs may also potentially serve as clinical biomarkers to stratify patients and investigate symptom-related molecular mechanisms. For example, transfer RNA fragments (tRFs) share several similarities with miRNAs in terms of their biological functions and molecular structure [68,69]. Both tRFs and miRNAs regulate gene expression at the post-transcriptional level by binding to target mRNAs. Additionally, tRFs are expected to have high post-sampling stability due to their short RNA chain length, similar to miRNAs, making them attractive candidates as a potential clinical biomarker. Future research should consider including tRFs in analyses to provide a more comprehensive understanding of small RNA-mediated gene regulation in schizophrenia.

## 4. Materials and Methods

### 4.1. Study Population

This study was conducted as an exploratory investigation utilizing the samples in the biobank of National Center of Neurology and Psychiatry (NCNP, Tokyo, Japan), which is an ISO 20387 accredited biobank, and was approved by the Ethical Research Practice Committee of Daiichi Sankyo Co., Ltd. (Tokyo, Japan). Schizophrenia patients underwent a structured interview using the Mini-International Neuropsychiatric Interview (M.I.N.I) [70], Japanese version, administered by trained psychologists or psychiatrists. A consensus diagnosis was made according to the DSM-IV criteria on the basis of the M.I.N.I, additional unstructured interviews, and information from medical records. Patients with a history of central nervous system disease or severe head injury were excluded. Seventy patients were assessed by PANSS [5]. Forty-one patients out of the seventy patients were assessed using the Japanese version of the BACS [71]. BACS composite scores were calculated by averaging the *z*-scores of the six subtests based on the mean and standard deviation from a healthy control group for each age group and sex [72].

### 4.2. Plasma Sample Preparation

Blood samples were collected from the patients within 67 days (mean 2.6 days) of the PANSS assessment and 412 days (mean 5.9 days) of the BACS assessment. The blood sample collection was performed via venipuncture into 7 mL EDTA-2Na-containing vacuum blood collection tubes (VENOJECT II, Terumo, Tokyo, Japan). The samples were centrifuged at 2500× *g* for 10 min at 4 °C, dispensed in screw-capped polypropylene tubes (96 Jacket Tubes 1.3 mL internal type, FCR&Bio Co., Ltd., Hyogo, Japan). The resulting plasma samples were stored in a deep freezer (−80 °C). The plasma samples were further centrifuged at 2500× *g* for 15 min at room temperature, and their supernatant (platelet-poor plasma) was used for RNA extraction.

### 4.3. MicroRNA Measurement

We measured miRNA levels in the plasma samples obtained from all 70 patients. RNA was extracted from the platelet-poor plasma samples using the Maxwell RSC miRNA Plasma and Serum kit and the Maxwell RSC system (Promega, Madison, WI, USA) in accordance with the manufacturer’s recommendation. Extracted RNAs were reverse transcribed to cDNAs by conformational restricted miRNA specific-RT primers and the ID3EAL cDNA Synthesis System (MiRXES Pte Ltd., Singapore). Expression levels of 376 miRNAs were quantified using MiRXES ID3EAL^TM^ PanoramiR miRNA assays (MiRXES Pte Ltd.) based on real-time polymerase chain reaction. The miRNA quantification results are presented as a raw threshold cycle (Ct value), which represents the polymerase chain reaction cycle upon reaching a designated threshold amplification level. The miRNAs of which amplification levels did not reach the designated threshold after 40-cycle amplification were considered below the limit of quantification and were regarded as 41-cycle amplification (Ct value = 41). We applied a global mean normalization to the miRNA Ct values using Equation (1), which is the same as in the previous study [27], because the global mean normalization outperforms the normalization using stable internal controls in terms of better reduction in technical variation and more accurate appreciation of biological changes [73].
(1)$$m_{i,j}=-(c_{i,j}-\frac{\sum_{i=1}^{i_{total}\;\;}c_{i,j}}{i_{total}}),$$
where $m_{i,j}$ is the normalized miRNA level of the i-th miRNA in the j-th sample,$c_{i,j}$ is the Ct value of the i-th miRNA in the j-th sample, $i_{total}$ is the total number of miRNAs (i.e., 376), and the negative sign outside parentheses converts the Ct values into the miRNA levels so that higher miRNA levels correspond to higher miRNA concentrations (otherwise, higher miRNA levels correspond to lower miRNA concentrations).

Hierarchical clustering analysis for the plasma samples on miRNA levels was performed using the unweighted pair group method with arithmetic mean method and Euclidean distance on Python 3.9.10 (Python Software Foundation, Wilmington, DE, USA). The schizophrenia patients were clustered into subgroups according to the characteristic patterns of miRNA levels. The heatmap was visualized using Microsoft^®^ Excel^®^ for Microsoft 365 MSO 16.0.13127.21490 (Microsoft, Redmond, WA, USA).

### 4.4. Correlation Analysis Between miRNAs and Symptom Scores

We evaluated the association between individual miRNAs and the symptom scores (i.e., PANSS total scores, PANSS positive subscales, PANSS negative subscales, PANSS general psychopathy subscales, and BACS composite scores) using univariate analysis. Pearson correlation coefficients were calculated to assess the relationship between each miRNA level and the symptom scores. The correlations were considered significant when the Benjamini–Hochberg corrected *p* was less than 0.05.

### 4.5. Extracting Symptom-Related miRNAs via Variable Selection in Multivariate Regression Models

We evaluated the association between multiple miRNAs and the symptom scores using multivariate analysis. Multivariate regression models were built by applying PLS to *z*-score normalized input and reference variables [29]. The input variables are the miRNA levels, while the reference variables are the observed symptom scores. The number of latent variables, which is the tuning parameter in PLS, was determined so that RMSECV in leave-one-out cross validation was minimized in a range from one to three. The optimal combination of miRNAs was determined by the forward variable selection [74], which is an iterative process that begins with an empty model and sequentially adds the input variable that improves RMSECV the most until no remaining variables improve RMSECV. We regarded the optimal combination of miRNAs as symptom-related miRNAs. The final model for each symptom was calibrated using the symptom-related miRNAs of all the samples to obtain regression coefficients.

### 4.6. MicroRNA Set Enrichment Analysis for Symptom-Related miRNAs

MicroRNA set enrichment analysis was performed using TAM 2.0 [65]. TAM 2.0 is a web-based tool that compares query miRNAs with reference miRNA sets to infer functional associations. These reference sets were derived from manual curation of over 9000 papers and were last updated in 2020. We used the symptom-related miRNAs as the query miRNAs and selected ‘Mask cancer-related terms’ and ‘Mask non-standard terms’ in the analysis settings. We filtered the enriched terms using the following acceptance criteria: (i) category is either ‘Tissue Specificity’, ‘Function’, or ‘Transcription Factor’; (ii) multiple miRNAs are mapped; and (iii) false discovery rate < 0.05 as statistical significance for the enrichment. TAM 2.0 automatically maps the query miRNAs to their corresponding miRNA genes, collapsing mature miRNA names (e.g., hsa-miR-144-5p to hsa-miR-144) including all available duplicated genes (e.g., hsa-miR-194-5p to both hsa-miR-194-1 and hsa-miR-194-2). As a result, the number of mapped miRNAs could be larger than the number of query miRNAs.

### 4.7. Literature-Based Text Mining for miRNA-Associated Proteins/Diseases

We conducted an unbiased quantification of publications that associate miRNAs with diseases or proteins using EmBiology (Elsevier B.V., Amsterdam, The Netherlands). EmBiology extracts biological relationships between a query entity and predefined entities through natural language processing for studies. The predefined entities are categorized into several groups such as ‘proteins’, which contains IL-6 and TNF, and ‘diseases’, which contains schizophrenia and inflammation. The literature text includes the full texts of approximately 1850 scientific journals (ca. 925 journals from Elsevier B.V. and ca. 925 from other publishers) and approximately 36 million abstracts, including the latest publications. For example, when ‘miR-123′ is given as the query entity, EmBiology identifies descriptions such as ‘transfection of miR-123 decreased IL-6′, thereby recognizing that miR-123 is associated with IL-6.

Given a set of symptom-related miRNAs as the query entities, we quantified the publications that associate each miRNA with the predefined entities using EmBiology, filtered the associated entities by the categories ‘diseases’ or ‘proteins’, and reported the total number of the publications for all the miRNAs per each category. For the disease, oncology-related terms (e.g., ‘neoplasm’ and ‘metastasis’) were excluded. The text mining analysis was conducted in June 2024 to include recent research findings (especially compared with TAM2.0, which is based on data up to 2020).

### 4.8. Literature-Based Inference on Regulatory Functions of miRNAs on Pro-Inflammatory Cytokines

We explored experimental evidence on the regulatory functions (i.e., upregulation or downregulation) of miRNAs on representative pro-inflammatory cytokines (i.e., IL-1β, IL-6, and TNFα) through a manual literature survey. We focused on the experimental evidence supporting the hypothesis that exacerbated symptoms are associated with high inflammation. We considered experimental data from the literature as experimental evidence if the experiments involve intervention with a specific miRNA (e.g., transfection of an agonistic miRNA or a miRNA inhibitor) and measurement of the cytokines (e.g., protein levels of IL-1β by enzyme-linked immunosorbent assay) in any species and any tissue. We accepted the regulatory functions of miRNAs on the cytokines as hypothetical pathways if these functions were demonstrated in at least two independent experimental studies.

## 5. Conclusions

MicroRNAs are a potential biomarker to identify patient subgroups reflecting pathophysiological conditions and to investigate symptom-related molecular mechanisms in schizophrenia. Patient-derived miRNAs offer promising potential for advancing precision medicine by enabling patient stratification in clinical settings and enhancing our understanding of symptom-related pathogenesis, thereby facilitating both personalized treatment strategies and targeted drug development.

## Figures and Tables

**Figure 1 ijms-25-13522-f001:**
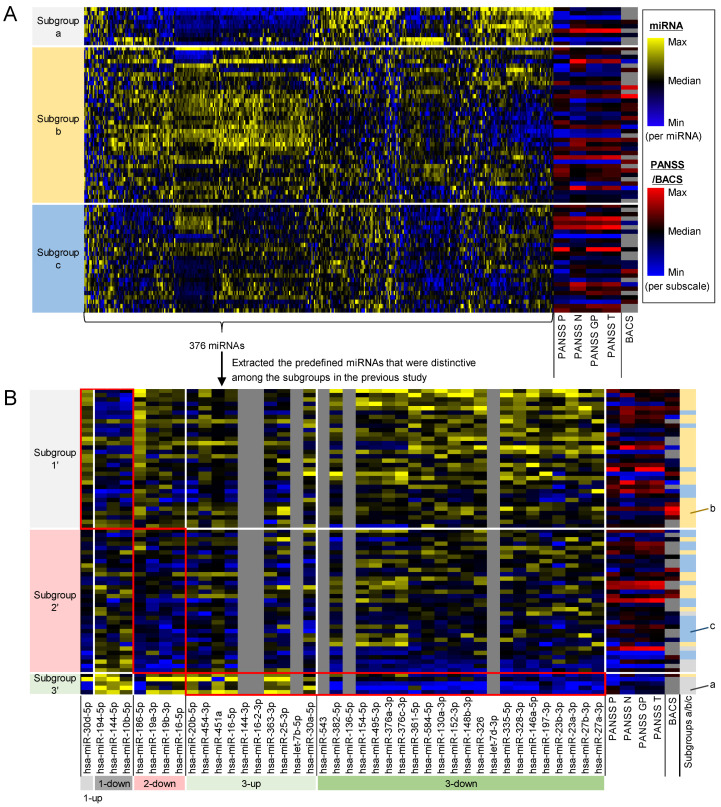
Plasma miRNA profiles identified similar subgroups of schizophrenia patients as in the previous study. (**A**) Whole miRNA profiles identified three subgroups of schizophrenia patients (*a*, *b*, and *c*). Heatmap shows expression levels of all the 376 miRNAs and scores of PANSS positive symptom subscales (PANSS P), PANSS negative symptom subscales (PANSS N), PANSS general psychopathy subscales (PANSS GP), PANSS total (PANSS T), and BACS composite (BACS). BACS scores in 29 out of 70 patients were not measured (gray-colored cells). (**B**) The patient subgroups based on the whole miRNAs were similar to those based on the predefined 40 miRNAs that were distinctive in the subgroups of the previous study. Heatmap shows expression levels of the predefined 40 miRNAs extracted from the whole 376 miRNAs, and 6 out of the predefined 40 miRNAs were not measured in the measurement platform (gray-colored cells). The predefined miRNAs also identified three subgroups of patients (1′, 2′, and 3′). Red frames indicate the miRNAs that were distinctive in each subgroup of the previous study. ‘1-up’ and ‘1-down’ are the upregulated and downregulated miRNAs in the subgroup 1 of the previous study, respectively. ‘2-down’ is the downregulated miRNAs in the subgroup 2 of the previous study. ‘3-up’ and ‘3-down’ are the upregulated and downregulated miRNAs in the subgroup 3 of the previous study, respectively.

**Figure 2 ijms-25-13522-f002:**
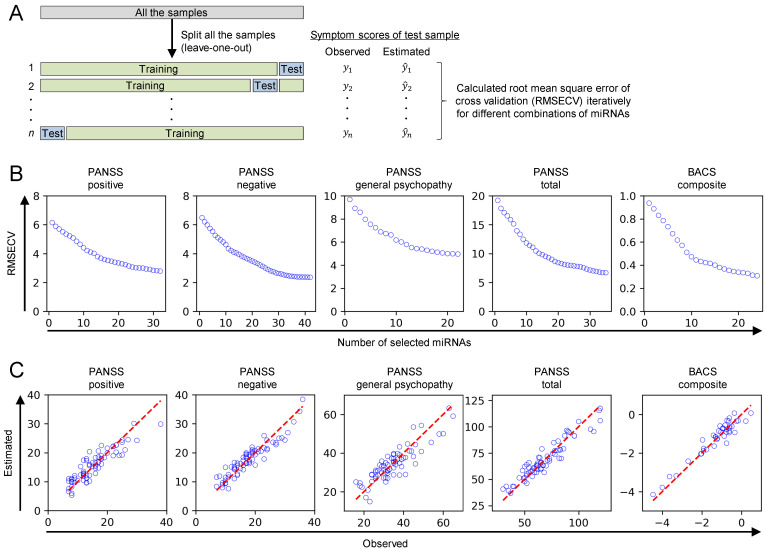
Multivariate analysis identified symptom-related miRNAs as optimal input variables to estimate symptom scores. (**A**) The scheme explains leave-one-out cross validation and variable selection. (**B**) The number of miRNAs as input variables was increased until no remaining variables that improve RMSECV. Forward stepwise method was used as the variable selection method. (**C**) The estimation accuracy of the models with optimal miRNAs was confirmed by scatter plot between observed and estimated symptom scores in leave-one-out cross validation. The red dashed lines represent the line of perfect agreement between observed and estimated symptom scores.

**Figure 3 ijms-25-13522-f003:**
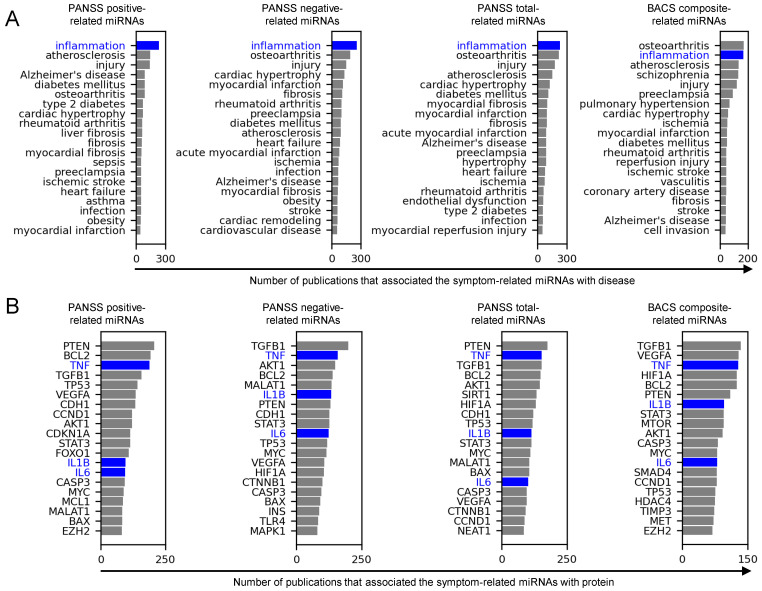
Symptom-related miRNAs were frequently associated with inflammation, IL-1β, IL-6, and TNFα in literature text. (**A**) Bar charts show the total number of publications that associated the symptom-related miRNAs with the predefined entities within the ‘disease’ category. Blue color highlights inflammation. (**B**) Bar charts show the total number of publications that associated the symptom-related miRNAs with the predefined entities within the ‘protein’ category. Blue color highlights IL-1β, IL-6, and TNFα.

**Figure 4 ijms-25-13522-f004:**
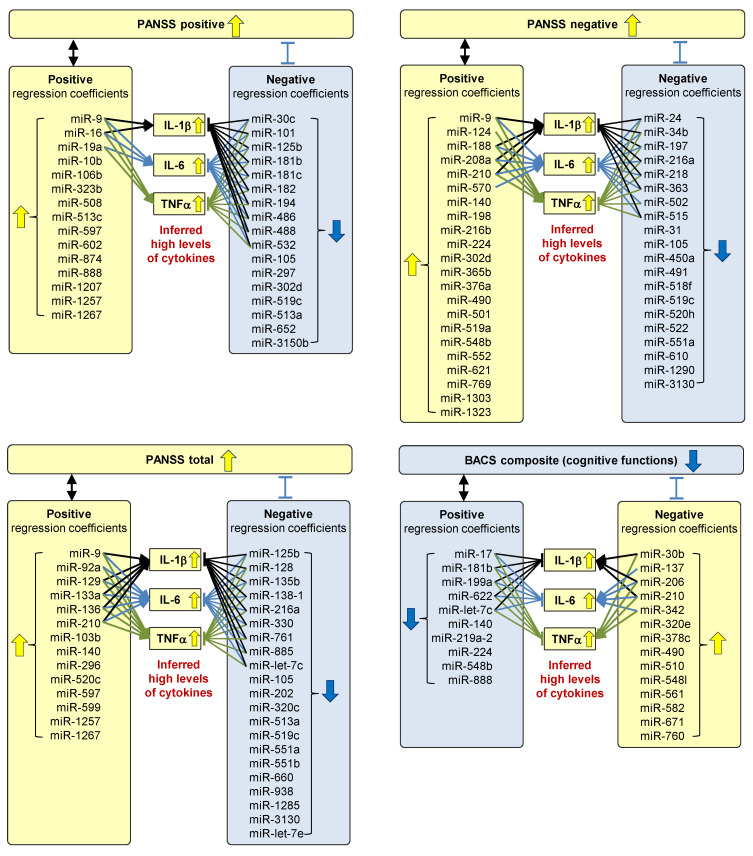
Patients with severe symptom scores were inferred to have high levels of IL-1β, IL-6, and TNFα. Each input variable (i.e., miRNA) in the models estimating the symptom scores (PANSS positive subscales, PANSS negative subscales, PANSS total scores, and BACS composite scores) has either positive or negative regression coefficient. High symptom scores were associated with high levels of miRNAs with positive regression coefficients and low levels of miRNAs with negative regression coefficients. Several miRNAs were reported to upregulate (indicated by a standard arrow) or downregulate (indicated by a flat-headed arrow) IL-1β, IL-6, and TNFα. High levels of miRNAs that upregulate these cytokines and low levels of miRNAs that downregulate these cytokines were associated with high cytokine levels. Combining these relationships, exacerbated symptoms (i.e., high PANSS scores and low BACS scores) were associated with high levels of these cytokines.

**Table 1 ijms-25-13522-t001:** Characteristics of schizophrenia patients.

Characteristics	*n* = 70 (Overall)	*n* = 41 (with BACS)
Age, mean ± S.D.	35.0 ± 13.7	31.2 ± 14.7
Male, *n* (%)	29 (41.4)	18 (43.9)
Race: Japanese, *n* (%)	70 (100)	41 (100)
PANSS total score, mean ± S.D.	68.4 ± 20.7	68.0 ± 18.1
PANSS positive symptom subscale score, mean ± S.D.	15.8 ± 6.5	15.4 ± 6.4
PANSS negative symptom subscale score, mean ± S.D.	17.9 ± 6.8	18.2 ± 6.8
PANSS general psychopathy subscale score, mean ± S.D.	34.8 ± 10.3	34.4 ± 8.8
BACS composite score, mean ± S.D.	-	−1.27 ± 1.08

**Table 2 ijms-25-13522-t002:** Correlation coefficients between miRNA levels and symptom scores.

Symptom Score	miRNA	*r*	*q*-Value
PANSS positive subscale	hsa-miR-519c-3p	−0.405	0.322
	hsa-miR-650	−0.349	0.819
	hsa-miR-1296-5p	0.290	0.838
PANSS negative subscale	hsa-miR-208a-3p	0.339	0.838
	hsa-miR-518f-3p	−0.334	0.838
	hsa-miR-27b-3p	0.326	0.838
PANSS general psychopathy subscale	hsa-miR-519c-3p	−0.390	0.396
	hsa-miR-551a	−0.364	0.740
	hsa-miR-34b-5p	−0.289	0.838
PANSS total	hsa-miR-519c-3p	−0.419	0.291
	hsa-miR-551a	−0.327	0.838
	hsa-miR-150-5p	−0.296	0.838
BACS composite	hsa-miR-320e	−0.536	0.291
	hsa-miR-671-5p	−0.458	0.819
	hsa-miR-500b-5p	0.444	0.838

*r*, Pearson’s correlation coefficient. *q*-value, Benjamini–Hochberg corrected *p*-value. The most significant three correlations per symptom score in terms of q-values were displayed although no single combination showed significant correlation (*q*-value > 0.05).

**Table 3 ijms-25-13522-t003:** Enriched pathways by symptom-related miRNAs.

miRNA Set	Enriched Pathway	*N*_mapped_/*N*_predefined_	FDR
PANSS positive-related miRNAs (32 miRNAs)	Immune Response	18/92	6.24 × 10^−10^
	Inflammation	15/112	9.25 × 10^−6^
	Cell Differentiation	11/56	9.82 × 10^−6^
	NFKB1	8/26	1.87 × 10^−5^
	Cell Death	11/78	2.33 × 10^−4^
	Brain.cerebellum	6/21	8.43 × 10^−4^
	Aging	9/63	1.31 × 10^−3^
	Plasma Cell Differentiation	4/8	1.71 × 10^−3^
	Hormone-mediated Signaling Pathway	8/58	2.81 × 10^−3^
	DNA Damage Repair	5/19	3.15 × 10^−3^
	Embryonic Stem Cell Differentiation	6/31	3.44 × 10^−3^
	Cell Motility	5/21	4.90 × 10^−3^
	Cell Cycle	9/83	5.20 × 10^−3^
	Circadian Rhythm	5/22	5.50 × 10^−3^
	STAT3	3/5	5.82 × 10^−3^
	Megakaryocyte Differentiation	3/5	5.82 × 10^−3^
	Apoptosis	10/106	6.01 × 10^−3^
	Brain Development	6/36	6.50 × 10^−3^
	Neuron Differentiation	4/14	8.52 × 10^−3^
	AKT1	3/6	9.33 × 10^−3^
	Hematopoiesis	7/57	1.08 × 10^−2^
	Response to Estrogen	3/8	1.77 × 10^−2^
	MYC	5/40	4.84 × 10^−2^
PANSS negative-related miRNAs (42 miRNAs)	Neuron Differentiation	9/14	3.26 × 10^−8^
	Inflammation	16/112	3.14 × 10^−5^
	Testis	11/47	4.39 × 10^−5^
	HIF1A	6/11	1.03 × 10^−4^
	Cell Differentiation	11/56	1.23 × 10^−4^
	Apoptosis	14/106	3.57 × 10^−4^
	Cell Proliferation	12/80	4.53 × 10^−4^
	Embryonic Stem Cell Differentiation	8/31	4.68 × 10^−4^
	Cell Cycle	12/83	6.04 × 10^−4^
	Hormone-mediated Signaling Pathway	10/58	8.02 × 10^−4^
	Brain Development	8/36	1.02 × 10^−3^
	Neuron Apoptosis	5/15	4.58 × 10^−3^
	Placenta	4/11	1.23 × 10^−2^
	Adipogenesis	5/20	1.25 × 10^−2^
	Brain	5/20	1.25 × 10^−2^
	Osteogenesis	8/59	1.33 × 10^−2^
	Muscle Development	5/21	1.39 × 10^−2^
	NFKB1	5/26	3.02 × 10^−2^
	BMP2	2/2	3.02 × 10^−2^
	Embryonic Development	4/17	3.62 × 10^−2^
	Smooth Muscle Cell Proliferation	4/18	4.48 × 10^−2^
	Erythrocyte Differentiation	3/9	4.89 × 10^−2^
PANSS general psychopathy-related miRNAs (22 miRNAs)	No item was significantly enriched	**-**	**-**
PANSS total-related miRNAs (35 miRNAs)	Brain	8/20	1.48 × 10^−5^
	Testis	11/47	2.87 × 10^−5^
	Embryonic Stem Cell Differentiation	9/31	2.96 × 10^−5^
	Cell Death	12/78	1.22 × 10^−4^
	Cell Proliferation	12/80	1.26 × 10^−4^
	Immune Response	13/92	1.45 × 10^−4^
	Cell Cycle	12/83	1.72 × 10^−4^
	Hormone-mediated Signaling Pathway	10/58	2.78 × 10^−4^
	Brain Development	8/36	4.27 × 10^−4^
	Inflammation	13/112	5.72 × 10^−4^
	Cell Differentiation	9/56	1.11 × 10^−3^
	Apoptosis	12/106	1.30 × 10^−3^
	MYC	7/40	4.47 × 10^−3^
	Glucose Metabolism	6/28	4.56 × 10^−3^
	Hematopoiesis	8/57	5.48 × 10^−3^
	HIF1A	4/11	6.89 × 10^−3^
	STAT3	3/5	8.54 × 10^−3^
	Cell Motility	5/21	8.61 × 10^−3^
	Wound Healing	5/23	1.20 × 10^−2^
	AKT1	3/6	1.38 × 10^−2^
	NFKB1	5/26	1.79 × 10^−2^
	Innate Immunity	6/42	1.95 × 10^−2^
	Cardiogenesis	3/7	2.02 × 10^−2^
	Keratinocyte Proliferation	2/2	2.18 × 10^−2^
	Latent Virus Replication	4/17	2.30 × 10^−2^
	Aging	7/63	2.99 × 10^−2^
	Angiogenesis	7/65	3.47 × 10^−2^
BACS composite-related miRNAs (24 miRNAs)	Inflammation	12/112	2.03 × 10^−5^
	Cell Proliferation	9/80	7.91 × 10^−4^
	Immune Response	8/92	7.56 × 10^−3^
	Myocardium	6/41	1.06 × 10^−2^
	Cell Death	7/78	1.27 × 10^−2^
	Osteogenesis	6/59	1.78 × 10^−2^
	Brain.nucleus_caudatus	3/10	3.55 × 10^−2^
	NFKB1	4/26	4.20 × 10^−2^

*N*_mapped_, number of mapped miRNAs; *N*_predefined_, number of predefined miRNAs in each pathway; FDR, false discovery rate. A list of the symptom-related miRNAs is shown in Appendix A.

## Data Availability

The data presented in this study are available on request from the corresponding author.

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
