# Peer review of "Plasma microRNAs Associate Positive, Negative, and Cognitive Symptoms with Inflammation in Schizophrenia"

_ijms, 2024, doi:10.3390/ijms252413522_

Round 1

Reviewer 1 Report

Comments and Suggestions for Authors

1. This interesting study explores the relevance of plasma microRNAs (miRs) to the severity of symptoms in schizophrenia patients by extending a previous effort in this direction by the same team of authors. The study non-surprisingly identified relevance of miRs differences in the patients’ plasma to their symptoms severity and to inflammation control; but it did not refer to transfer RNA fragments (tRFs) whose activities are largely shared with those of miRs and may add to the reported symptoms and their severity, while limiting the repetitiveness of the conveyed results.

2. Additionally, the authors keep mentioning exosomes as potentially relevant to the symptoms’ severity, but to the best of my understanding this contribution emerged as only adding a minor impact, if any to these observations. The authors may add more information if they wish to keep this part of the manuscript as is.

3. However, there is no reference in this study to the impact of age on the observed results, although age-related issues are well documented in the studied diseases.

4. Last, but not least the impact of race may have a profound effect on mental patterns which should be dealt with in the revised manuscript.

Reviewer 2 Report

Comments and Suggestions for Authors

In the article entitled “Plasma microRNAs associate positive, negative, and cognitive symptoms with inflammation in schizophrenia” the authors study miRNA profiles in plasma from patients with schizophrenia to understand the relationship between inflammation and positive, negative, and cognitive symptoms of the disease. The authors evaluated 376 miRNAs in plasma samples from patients and correlated them with scores on scales such as PANSS and BACS.

It is an interesting article as they do a stratification of patients based on miRNA profiles. They also made an association of positive, negative and cognitive symptoms associated with NF-kB and proinflammatory cytokines.

There are some limitations that need to be addressed or at least taken into consideration such as:

1.   Sample size and demographic factors such as ethnicity.

2.   Accuracy in the model’s estimation of symptom scores, where it was only evaluated by cross-validation within the dataset, which does not guarantee that the selected miRNAs are not over-fitted to the obtained dataset. It is suggested to test models on independent data sets.

3.   Do a longitudinal analysis, which could provide information on whether the profiles reflect stable characteristics or could vary.

4.         Apparently, other variables such as drug treatment history, diet, etc., were not controlled.

Round 2

Reviewer 1 Report

Comments and Suggestions for Authors

When a serious journal tells the authors to subject their manuscript to a major revision, this request is often accompanied by a statement of how many months may be devoted to such a revision. Here, the authors ran a quick revision which does not change the value of their manuscript and completed it within one week. This seems to reflect low appreciation of the value of their own work. What is needed here is an in-depth revision, such that if the authors refer to relevance of transfer RNA fragments (tRFs) to the phenomenon they describe, they should refer to this issue in the introduction and methods, analyze each of the bulk tRFs whose levels were changed individually and refer to their properties both in the introduction and the results and discussion and cite relevant articles by others on such changes.

Author Response

We appreciate the reviewer's thorough comments and understand the concern regarding the speed of our revision. We would like to clarify that our quick response does not reflect a lack of appreciation for our work or the review process, but rather our commitment to addressing the comments in accordance with the journal's requirement for authors to submit the revisions within 10 days.

We agree that analyzing transfer RNA fragments (tRFs) would provide additional valuable insights. However, we regret that we cannot perform additional experiments on tRFs due to the limited availability of the samples used in this study.

We respectfully maintain that the focus of this study is on miRNAs, and the results and discussion presented are valid and significant even without tRF data. The current findings on miRNAs contribute meaningful insights to the field of schizophrenia research.

As mentioned in the limitations section added during the previous revision process, we acknowledge the potential importance of tRFs and suggest that future research should evaluate tRFs to provide a more comprehensive understanding of small RNA-mediated gene regulation in schizophrenia.

We believe that our current manuscript, with its focus on miRNAs, presents a complete and valuable contribution to the field, while also recognizing areas for future exploration.

Reviewer 2 Report

Comments and Suggestions for Authors

Thanks to the authors for their responses. From my point of view, changes are no longer required.

Author Response

We sincerely thank the reviewer for confirming that we have responded to the comments appropriately. 

Round 3

Reviewer 1 Report

Comments and Suggestions for Authors

The Authors are requested to prepare a major revision as per my previous comments.